# Strong evidence for the adaptive walk model of gene evolution in *Drosophila* and *Arabidopsis*

Ana Filipa Moutinho[1,2¤]*, Adam Eyre-Walker[2], Julien Y. Dutheil[1,3]

**1** Department of Evolutionary Genetics, Max Planck Institute for Evolutionary Biology, Plön, Germany,
**2** School of Life Sciences, University of Sussex, Brighton, United Kingdom, **3** Unité Mixte de Recherche 5554 Institut des Sciences de l'Evolution, CNRS, IRD, EPHE, Université de Montpellier, Montpellier, France

¤ Current address: School of Life Sciences, University of Sussex, Brighton, United Kingdom
* a.f.moutinho@sussex.ac.uk

**Data Availability Statement:** All data and detailed R scripts necessary to reproduce the results are available in the supplementary data online, https://gitlab.gwdg.de/molsysevol/supplementarydata_geneage and https://zenodo.org/record/6828430.

## Abstract

Understanding the dynamics of species adaptation to their environments has long been a central focus of the study of evolution. Theories of adaptation propose that populations evolve by "walking" in a fitness landscape. This "adaptive walk" is characterised by a pattern of diminishing returns, where populations further away from their fitness optimum take larger steps than those closer to their optimal conditions. Hence, we expect young genes to evolve faster and experience mutations with stronger fitness effects than older genes because they are further away from their fitness optimum. Testing this hypothesis, however, constitutes an arduous task. Young genes are small, encode proteins with a higher degree of intrinsic disorder, are expressed at lower levels, and are involved in species-specific adaptations. Since all these factors lead to increased protein evolutionary rates, they could be masking the effect of gene age. While controlling for these factors, we used population genomic data sets of *Arabidopsis* and *Drosophila* and estimated the rate of adaptive substitutions across genes from different phylostrata. We found that a gene's evolutionary age significantly impacts the molecular rate of adaptation. Moreover, we observed that substitutions in young genes tend to have larger physicochemical effects. Our study, therefore, provides strong evidence that molecular evolution follows an adaptive walk model across a large evolutionary timescale.

## Introduction

How does adaptive evolution proceed in space and in time? This question has long intrigued evolutionary biologists. Fisher [1] proposed that adaptation relies on mutations with small effect sizes at the phenotypic level. He presented the geometric model of adaptation where phenotypic evolution occurs continuously and gradually toward an optimum fitness [1]. At the molecular level, Wright [2,3] first introduced the idea that populations evolve in the space of all possible gene combinations to acquire higher fitness. He characterised this model of evolution as a "walk" in an adaptive landscape. Wright consequently proposed the shifting balance

**Funding:** JYD received funding from the Max Planck Society. The funders had no role in study design, data collection and analysis, decision to publish, or preparation of the manuscript.

**Competing interests:** The authors have declared that no competing interests exist.

**Abbreviations:** ANCOVA, analysis of covariance; DFE, distribution of fitness effects; GO, gene ontology; MK, MacDonald–Kreitman; PPI, protein–protein interaction; RSA, relative solvent accessibility; SFS, site frequency spectrum.

theory of adaptation, which combines the effects of drift and selection. Drift acts by moving the population away from its local peak, while natural selection directs the population to higher fitness, the so-called "global optimum" in the fitness landscape. With the rise of molecular genetics, Maynard Smith [4] extended this idea to a sequence-based model of adaptation. He introduced the idea of an "adaptive walk," where a protein "walks" in the space of all possible amino acid sequences towards the ones with increasingly higher fitness values. Wright's and Smith's adaptation model was further extended by Gillespie [5–7], who presented the "move rule" in an adaptive landscape. Following Kimura's work on the effect sizes of mutations [8], Gillespie suggested that adaptation proceeds in large steps, where mutations with higher fitness effects are more likely to reach fixation. The adaptive walk model was later fully developed by Allen Orr [9,10], who extended Fisher's geometric model of adaptation and showed that adaptive walks lead to a pattern of diminishing returns. A sequence further away from its local optimum tends to accumulate large-effect mutations at the beginning of the walk, small-effect mutations being then only fixed when the sequence is approaching its optimum fitness. Therefore, under this model, adaptation relies both on mutations of large and small fitness effects, in a time-dependent manner. Experimental studies tracing the evolution of bacteria [11–14] and fungi [15] have supported this model. Experimental studies, however, can only assess patterns of adaptation at relatively short time scales in artificial environments and are limited to certain organisms. The challenge lies in studying adaptation across long evolutionary time scales: How does the distribution of beneficial mutations vary over time?

While long-term evolutionary processes are not directly observable, they leave a signature in genome sequences. Species share genes to variable extents, according to their degree of divergence. Some ancient genes, which evolved early, are shared by many species. Others, which evolved more recently, are only shared by a few related species [16,17]. Consequently, the age of a gene can be inferred from its phyletic pattern, i.e., its presence or absence across the phylogeny [18]. These phyletic patterns are reconstructed using sequence similarity searches performed by tools like BLAST [19]. A gene is considered "old" if a homolog is identified in several taxa over a deep evolutionary scale, or "young" or lineage-specific if the recognised homologs are only present in closely related species. This approach is known as phylostratigraphy [20].

Previous studies suggested that young or lineage-specific protein-coding genes evolve faster than older ones [17,21–27]. Albà and Castresana [27] showed a negative correlation between the ratio of the nonsynonymous ($d_N$) to synonymous ($d_S$) substitutions rates, ω, and gene age in the divergence between humans and mice, with young genes presenting a higher ω. Similar correlations between ω and gene age have been observed in primates [22], fungi [25], *Drosophila* [23,28,29], bacteria [30], viruses [31], plants [32,33], and protozoan parasites [34]. Despite the observed consistency across taxa, the underlying causal mechanisms remain debated [21]. As the average $d_N/d_S$ ratio integrates both positive and negative selection, a comparatively higher ratio can result from a less stringent purifying selection, the occurrence of positive selection, or both. By looking at polymorphism data, Cai and Petrov [22] suggested that the faster evolution in young primate genes may be due to the lack of selective constraint posed by purifying selection, while analyses of adaptive evolution were inconclusive.

Analysing the effect of gene age on adaptive evolution is a complex task, as young and old genes differ in their structural properties, expression level, and protein function. Young genes tend to be smaller [22,24,35], have a higher degree of intrinsic disorder [36], and are expressed at lower levels [17,22,24,26]. Moreover, young genes tend to encode proteins involved in developing species-specific characteristics and immune and stress responses [16,37,38]. As the macromolecular structure [39,40], gene expression levels [39,41] and protein function [39,42,43] are known determinants of the rate of protein adaptation, they could be confounding the effect

of gene age. Several studies reported the substantial impact of gene expression on the adaptive rate of proteins, where highly expressed proteins are significantly more constrained and have lower adaptation rates [39,41,44,45]. At the macromolecular level, some studies showed that highly disordered [39,40] and exposed residues [39] present higher rates of adaptive evolution. Finally, there is evidence that proteins involved in the immune and stress response have higher molecular adaptive rates [39,43,46,47]. Thus, controlling for these confounding factors when assessing the impact of gene age on the rate of molecular adaptation is crucial.

Here, we used a population genomic approach to test the adaptive walk model. We make 2 predictions: first, that younger genes are undergoing faster rates of adaptive evolution, and second, the evolutionary steps they make are larger. We tested the first prediction by estimating rates of adaptive and nonadaptive protein evolution using an extension of the MacDonald–Kreitman (MK) test [48], which uses counts of polymorphism and substitution at selected and neutral sites. We quantified the rates of adaptive and nonadaptive evolution using the statistics $\omega_a$ and $\omega_{na}$, which denote the rates of adaptive and nonadaptive nonsynonymous substitution relative to the mutation rate. We investigated whether protein length, gene expression, relative solvent accessibility (RSA), intrinsic protein disorder, BLAST's false-negative rate, and protein function act as confounding factors of the effect of gene age. To test the second prediction, we considered the rates of substitution between amino acids separated by different physicochemical distances as a function of gene age. We tested our hypotheses in 2 pairs of species with different life history traits: the Diptera *Drosophila melanogaster* and *Drosophila simulans* and the Brassica *Arabidopsis thaliana* and *Arabidopsis lyrata*. In each species pair, we compared their most recent genes with those dating back to the origin of cellular organisms.

## Results

### Young genes have a higher rate of adaptive substitutions

We tested the adaptive walk model of sequence evolution by assessing the impact of gene age on the rate of adaptive ($\omega_a$) and nonadaptive ($\omega_{na}$) nonsynonymous substitutions. We used Grapes [48] to estimate $\omega_a$ and $\omega_{na}$, which accounts for segregating slightly deleterious and advantageous mutations by specifically modelling the distribution of fitness effects (DFE). Gene age data were obtained from published data sets [28,33]. We found that gene age is significantly correlated to estimates of $\omega$, $\omega_a$, and $\omega_{na}$ in both species' pairs (Table 1 and Fig 1B; data available in S1 Data). This result suggests that the higher $\omega$ ratio of more recently evolved genes is due to a higher rate of adaptive and nonadaptive nonsynonymous substitutions. As X-linked genes are known to evolve faster due to the male hemizygosity [49–51], we assessed whether the relationship between evolutionary rates and gene age differed between chromosomes in *Drosophila* (Fig 1B). We compared models with and without the chromosome's effect (see Material and methods and S1 File) and found only low support for a chromosomal effect ($p = 0.041$ for $\omega_{na}$ and $p = 0.094$ for $\omega_a$). We, therefore, combined data from all chromosomes for subsequent analyses.

### The effect of gene age on the rate of molecular adaptation is robust to multiple confounding factors

Genes of different ages intrinsically differ in their features [16,22,36]. As such traits significantly impact the rate of molecular evolution [39], they may be confounding the faster adaptive rates observed in young genes. Previous studies reported that younger genes encode shorter proteins [24,35,52] and are expressed at lower levels [17,22,24,26], a pattern that we also observed in our data set (gene age versus protein length for *D. melanogaster* and *A. thaliana*,

**Table 1. Kendall's correlation coefficients for the relationship between ω, $\omega_{na}$, and $\omega_a$ and gene age, for the analysis of gene age and the combined analyses of gene age with the respective cofactors: protein length, gene expression, protein intrinsic disorder, and the mean relative solvent accessibility per gene.** The combined probabilities for each cofactor within and across species are presented in the fields "Weighted Z" and "Weighted Z across species," respectively, for ω, $\omega_{na}$, and $\omega_a$.

| | | *Arabidopsis* | | | *Drosophila* | | | Weighted Z across species | | |
|---|---|---|---|---|---|---|---|---|---|---|
| | | ω | $\omega_{na}$ | $\omega_a$ | ω | $\omega_{na}$ | $\omega_a$ | ω | $\omega_{na}$ | $\omega_a$ |
| Gene Age | | 0.962 *** | 0.848 *** | 0.733 *** | 0.727 *** | 0.697 ** | 0.636 ** | | | |
| Protein Length | Long | 1.000 ** | 0.867 * | −0.200 | 0.867 * | 0.600 (.) | 0.867 * | $1.56 \times 10^{-4}$ *** | $7.71 \times 10^{-5}$ *** | $7.98 \times 10^{-3}$ ** |
| | Short | 1.000 ** | 0.867 * | 0.600 (.) | 0.733 * | 0.867 * | 0.467 | | | |
| | Weighted Z | $6.46 \times 10^{-4}$ *** | $1.61 \times 10^{-3}$ ** | 0.133 | $2.64 \times 10^{-3}$ ** | $5.29 \times 10^{-3}$ ** | 0.0105 * | | | |
| Gene Expression | High | 0.867 * | 0.867 * | 0.467 | 0.867 * | 1.000 ** | 0.600 (.) | $6.93 \times 10^{-5}$ *** | $6.89 \times 10^{-6}$ *** | $3.53 \times 10^{-3}$ ** |
| | Low | 0.867 * | 1.000 ** | 0.333 | 0.867 * | 0.733 * | 1.000 ** | | | |
| | Weighted Z | $1.51 \times 10^{-3}$ ** | $3.71 \times 10^{-4}$ *** | 0.186 | $1.09 \times 10^{-3}$ ** | $1.68 \times 10^{-3}$ ** | $2.24 \times 10^{-3}$ ** | | | |
| Protein Intrinsic Disorder | High | 1.000 *** | 0.939 *** | 0.636 ** | 0.670 ** | 0.303 | 0.515 * | $<2 \times 10^{-216}$ *** | $6.60 \times 10^{-6}$ *** | $2.53 \times 10^{-3}$ ** |
| | Low | 0.970 *** | 0.909 *** | 0.454 * | 0.630 ** | 0.576 ** | 0.273 | | | |
| | Weighted Z | $<2 \times 10^{-216}$ *** | $<2 \times 10^{-216}$ *** | $1.20 \times 10^{-3}$ ** | $3.85 \times 10^{-5}$ *** | $5.80 \times 10^{-3}$ ** | $4.18 \times 10^{-2}$ * | | | |
| Mean Relative Solvent Accessibility | High | 0.944 *** | 0.889 *** | 0.722 ** | 0.636 ** | 0.673 ** | 0.564 * | $1.00 \times 10^{-7}$ *** | $9.00 \times 10^{-7}$ *** | $1.37 \times 10^{-5}$ *** |
| | Low | 1.000 *** | 0.778 ** | 0.667 * | 0.636 ** | 0.491 * | 0.564 * | | | |
| | Weighted Z | $6.20 \times 10^{-6}$ *** | $1.41 \times 10^{-5}$ *** | $1.24 \times 10^{-3}$ ** | $3.67 \times 10^{-4}$ *** | $7.76 \times 10^{-4}$ *** | $1.55 \times 10^{-3}$ ** | | | |

For each variable, the correlation coefficient and the combined probabilities are shown with the respective significance (*$P < 0.05$; **$P < 0.01$; ***$P < 0.001$; "." $0.05 \leq P < 0.10$) for ω, $\omega_{na}$, and $\omega_a$ in *Arabidopsis* and *Drosophila*. As the effect of gene age was assessed by combining genes in each age class, these correlation coefficients do not measure the intrinsic gene-level strength of correlation between gene age and molecular rates of evolution.

respectively: Kendall's τ = −0.485, $p = 2.82 \times 10^{-2}$; τ = −0.848, $p = 1.06 \times 10^{-5}$, S1A Fig, data available in S2 Data; gene age versus gene expression for *D. melanogaster* and *A. thaliana*, respectively: τ = −0.595, $p = 7.35 \times 10^{-3}$; τ = −0.790, $p = 4.00 \times 10^{-5}$, S1B Fig, data available in S3 Data). As younger proteins are shorter than older ones, they have a higher proportion of exposed residues [39]: Gene age is significantly positively correlated with the average RSA per gene (τ = 0.636, $p = 3.98 \times 10^{-3}$; τ = 0.695, $p = 3.03 \times 10^{-4}$, for *D. melanogaster* and *A. thaliana*, respectively; S2A Fig, data available in S4 Data). Because exposed residues are more flexible [53], young genes tend to encode proteins with a higher degree of intrinsic disorder, a pattern previously reported in mice [36]. We confirmed this pattern in *D. melanogaster* (τ = 0.606, $p = 6.10 \times 10^{-3}$; S2B Fig) and *A. thaliana* (τ = 0.467, $p = 1.53 \times 10^{-2}$; S2B Fig, data available in S5 Data). Given that $\omega_a$ and $\omega_{na}$ are negatively correlated to length and gene expression and positively correlated to RSA and protein disorder [39], these could be driving the effect of gene age on $\omega_a$ and $\omega_{na}$; i.e., the correlation between $\omega_a$ and $\omega_{na}$ and age could be incidental.

To control for the impact of each confounding factor using Grapes, we split our data into 2 roughly equal-sized groups according to protein length, expression level, average RSA, and average intrinsic disorder and reran the analysis within the "high" and "low" groups, combining probabilities from the 2 analyses using the weighted Z-method [54]. As estimates of the rate of adaptive substitutions for a small number of genes exhibit large sampling variances [55,56], each putative confounding factor could only be tested separately. Some phylostrata were further combined when underrepresented in some gene categories (see Material and methods). We found that ω, $\omega_{na}$, and $\omega_a$ remain significantly correlated to gene age, except when controlling for protein length and gene expression for $\omega_a$ in *Arabidopsis* (Fig 2 and Table 1; data available in S6–S9 Data tables). This weaker effect may be a consequence of how

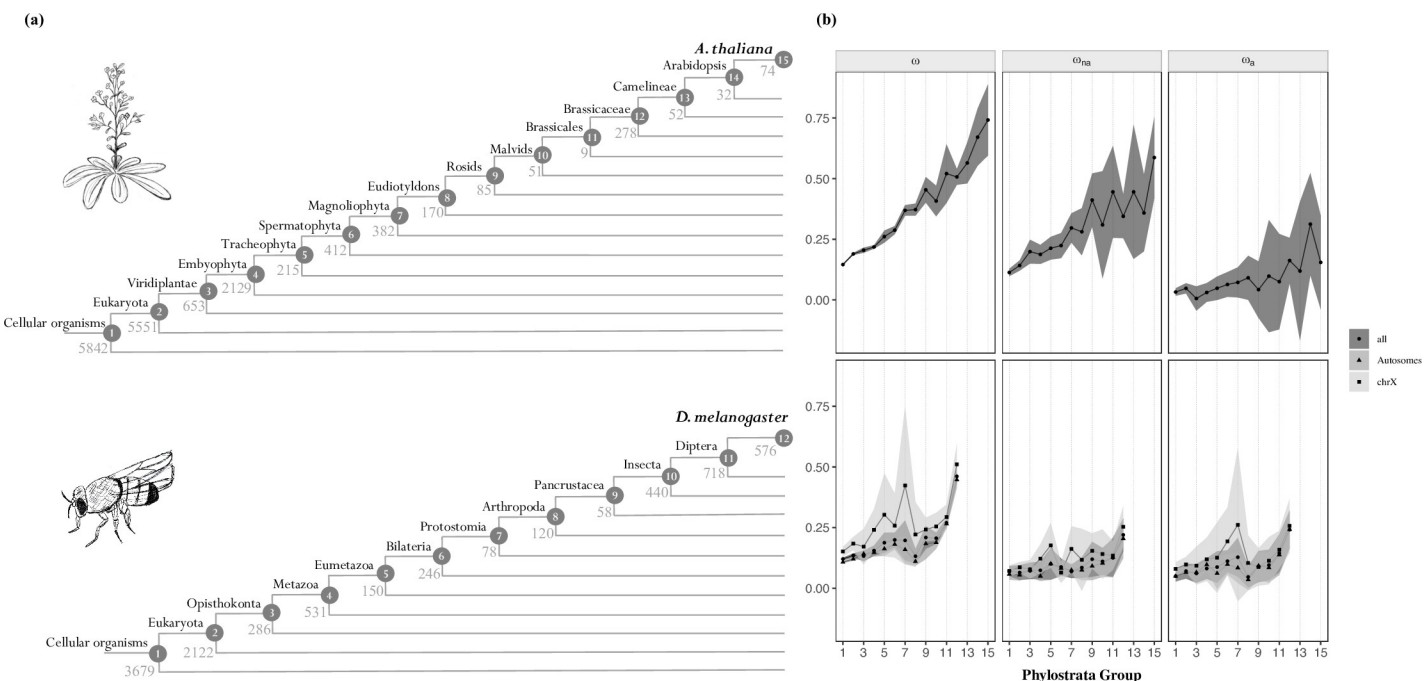

**Fig 1. (a)** Phylogenetic definition of the strata used in the analyses for *A. thaliana* (top) and *D. melanogaster* (bottom). The number of genes mapped to each clade is shown. **(b)** Relationship between the rate of protein evolution (ω), nonadaptive nonsynonymous substitutions ($\omega_{na}$), and adaptive nonsynonymous substitutions ($\omega_a$) with gene age in *A. thaliana* (top) and in *D. melanogaster* (bottom). Clades are ordered according to (a). In *D. melanogaster*, the results for X-linked, autosomal, and total genes are shown. Mean values of ω, $\omega_{na}$, and $\omega_a$ for each category are represented with the black points. Error bars denote for the 95% confidence interval for each category, computed over 100 bootstrap replicates. The data (S1 Data) and code needed to generate this table can be found at https://gitlab.gwdg.de/molsysevol/supplementarydata_geneage and https://zenodo.org/record/6828430.

the most recent clades were combined in these analyses, as there was little data available for those genes (see Material and methods). Nonetheless, when combining probabilities across the 2 species, we observed a significant correlation between all measures of evolutionary rate and gene age controlling each of the cofactors (Table 1; data available in S10–S13 Data tables).

We next asked whether the observed effect could be explained by the residual correlation of the cofactors with gene age within each "high" and "low" group. We performed 3 different tests: (1) the correlation between $\omega_a$ and the mean value of each cofactor in each age class within the "high" and "low" groups (data available in S14–S17 Data tables); (2) the correlation between the cofactor and gene age in each "high" and "low" group (data available in S18 Data); and (3) a linear model where $\omega_a$ is the response variable, and gene age, category, species (*Arabidopsis* or *Drosophila*), and the within category cofactor values are explanatory variables (data available in S14–S17 Data tables). While we do observe a general correlation of the cofactor with gene age within each group, we found that nearly all correlations between $\omega_a$ and the cofactor are nonsignificant (S1 Table). Moreover, the linear model analyses showed that in all cases but gene length, gene age is the most significant variable, with an effect consistent with that observed in Table 1. For gene length, gene age was only significant as an interaction with the species variable, having a positive effect only in *Drosophila* (S2 Table). These findings, therefore, suggest that the effect of gene age on $\omega_a$ is independent of the cofactor.

To jointly estimate the effect of the potential confounding factors, we applied a recently developed method that extends the MK test with a generalized linear model [57]. This approach disentangles the effects of each factor on the rate of adaptive substitutions per nucleotide site. However, this method does not model the DFE and hence cannot account for

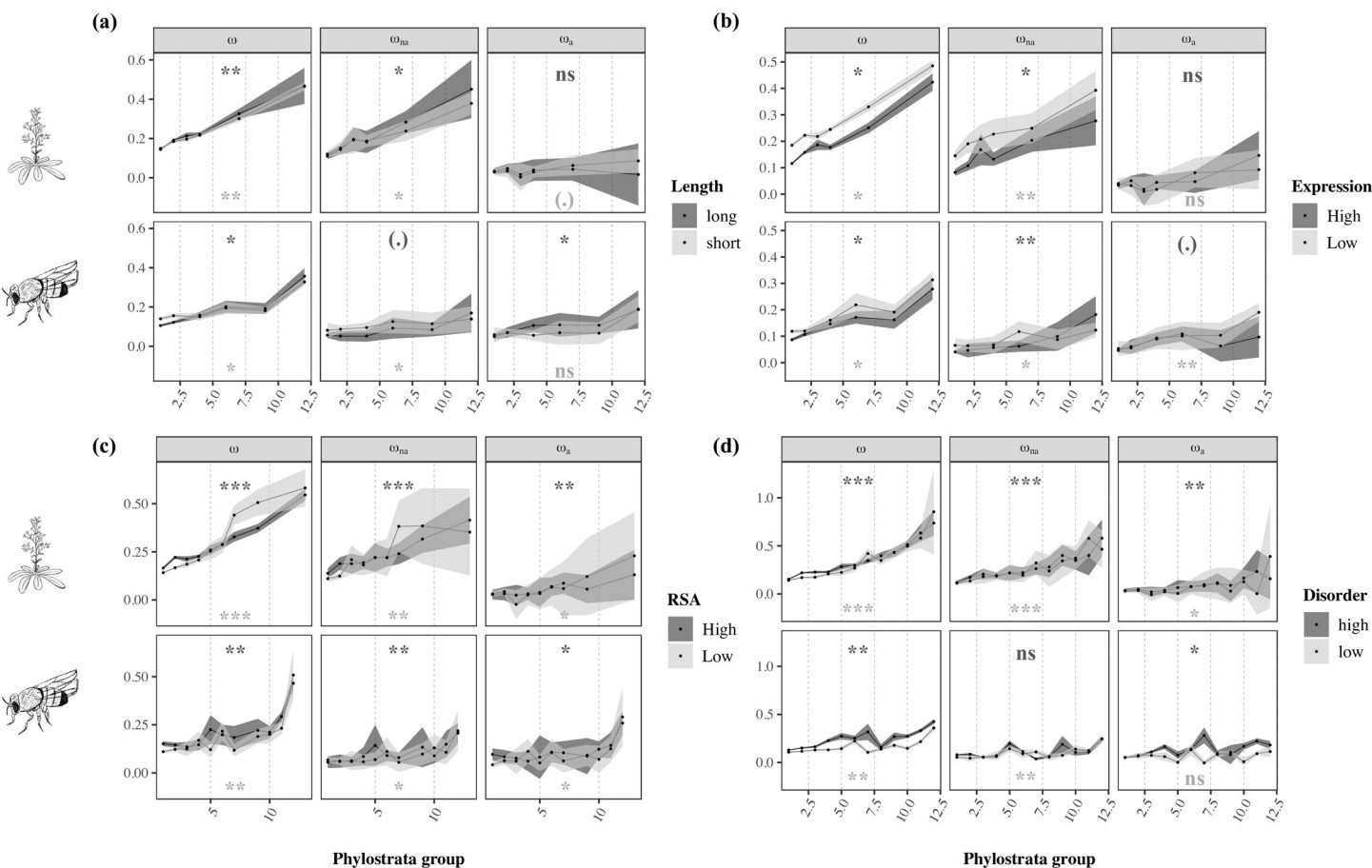

**Fig 2.** Estimates of ω, $ω_{na}$, and $ω_a$ plotted as a function of **(a)** protein length and **(b)** mean expression levels, **(c)** RSA, and **(d)** protein intrinsic disorder with gene age in *A. thaliana* (top) and *D. melanogaster* (bottom). Analyses were performed by comparing short and long genes **(a)**, lowly and highly expressed genes **(b)**, proteins with low and high mean RSA values **(c)**, and proteins with low and high average intrinsic disorder **(d)** across age categories (see Material and methods). Legend as in Fig 1. Significance levels are shown for each correlation between gene age and the rates of protein evolution (*$P < 0.05$; **$P < 0.01$; ***$P < 0.001$; "." $0.05 \leq P < 0.10$; "ns" $P > 0.10$). The data (S6–S9 Data tables) and code needed to generate this table can be found at https://gitlab.gwdg.de/molsysevol/supplementarydata_geneage and https://zenodo.org/record/6828430.

segregating slightly deleterious mutations, which can bias estimates of the rate of adaptive substitutions [58]. Hence, following the approach suggested in Huang [57], we removed sites for which the derived allele frequency was below 50% to minimize any potential bias. Despite the large reduction in the data set, this analysis revealed a significant effect of gene age (S3 Table; data available in S19–S22 Data tables). Our findings, therefore, suggest that the effect of gene age on rates of protein evolution is robust to the tested confounding factors and that a gene's age acts as a significant determinant of the rate of adaptive and nonadaptive evolution in both species.

Lastly, we aimed at assessing the effect size of gene age on $ω_a$ relative to other factors. Because correlation coefficients were computed from values averaged over multiple genes and genes were categorized differently for each analysis, the comparison of correlation coefficients does not provide a reliable estimate of relative effect sizes. Therefore, we assessed the relative contribution of each variable to the MK regression model by computing likelihood pseudo-$R^2$s [59]. These were estimated by comparing the log-likelihood of the full model (all factors included) with the log-likelihood of reduced models, where each single factor was removed independently from the full model. These estimates suggest that, despite its very significant

effect, gene age contributes comparatively little to the explained variance compared to factors such as protein function (S4 Table). To note, however, that these estimates refer to the relative contribution of each variable per nucleotide site and not per gene as this is the unit used by the MK regression, which results in generally low $R^2$ values.

## The effect of gene age on the molecular rate of adaptation is robust to BLAST's false negative rates

The phylostratigraphy approach has been previously used to date the emergence of new genes, and some studies have pointed out its potential limitations [28,60–63]. Because BLAST homology searches might fail to identify homologs in short or rapidly evolving genes, such genes could be mistakenly classified as young. As expected, we observed that genes in younger phylostrata present higher E-values in both species ($\tau = 0.564$, $p = 0.025$; $\tau = 0.951$, p = $1.20 \times 10^{-6}$, for *D. melanogaster* and *A. thaliana*, respectively; S3A Fig; data available in S23 Data). To control this effect, we reran our analyses using genes with very low E-values that are likely to be detected in most age strata and for which correlation between E-values and gene age was no longer significant (see Material and methods and S2 File) ($\tau = 0.408$, $p = 0.111$; $\tau = 0.354$, $p = 0.141$, for *D. melanogaster* and *A. thaliana*, respectively; S3B Fig; data available in S23 Data). We observed that the effect of gene age remained for all estimates in the 2 species when using this restricted data set ($\omega$: $\tau = 0.929$, p = $1.30 \times 10^{-3}$; $\omega_{na}$: $\tau = 0.786$, $p = 6.49 \times 10^{-3}$; $\omega_a$: $\tau = 0.643$, $p = 2.59 \times 10^{-2}$ in *A. thaliana*; and $\omega$: $\tau = 0.697$, $p = 1.61 \times 10^{-3}$; $\omega_{na}$: $\tau = 0.636$, $p = 3.98 \times 10^{-3}$; $\omega_a$: $\tau = 0.636$, $p = 3.98 \times 10^{-3}$ in *D. melanogaster*, S4 Fig; data available in S24 Data). These results suggest that the correlation of gene age with the rate of adaptive evolution cannot be attributed to errors in dating the emergence of a gene stemming from the failure of identifying homologs in older taxa.

## The effect of gene age on the rate of molecular adaptation does not depend on protein function

Lineage-specific genes are known to be involved in species-specific adaptive processes, such as the evolution of morphological diversity [64] and immune and stress responses [17,34,64]. As proteins encoding such functions tend to have higher molecular rates of adaptation [39,42,43,46,47,65], we further assessed whether the observed effect of gene age could be due to younger genes being enriched in functions with higher evolutionary rates. We first examined which functions are encoded by young genes in *A. thaliana* and *D. melanogaster*. In *A. thaliana*, young genes (clades 12 to 15 in Fig 1A) are mostly involved in a large variety of cellular processes, stress response and external stimulus, protein binding, and signal transduction (S5A Fig; data available in S25 Data). In *D. melanogaster*, young genes (clades 11 and 12 in Fig 1A) encode mostly functions involved in the cell's anatomic structure, stress response, nervous system processes, enzyme regulators, immune system mechanisms, and a wide range of metabolic processes (S5B Fig; data available in S25 Data).

To correct for the potential bias of protein function in the relationship between the rate of adaptive evolution and gene age, we split the genes into different functional categories based on gene ontology (GO) terms. To simultaneously control for the effect of gene age, we further divided the genes into 3 age categories, trying to keep a similar number of genes in each (see Material and methods). Grapes was then used to estimate $\omega$, $\omega_{na}$, and $\omega_a$ in each combined category. As some gene functions were biased towards some age categories (S5 Fig), we could not do this analysis for all GO terms. We, therefore, only used the GO terms with a sufficient number of annotated genes in each age class (see Material and methods). In *A. thaliana*, we found that the impact of gene age on $\omega_a$ is stronger in proteins linked to stress response and cellular

components, where younger genes present higher molecular adaptive rates (S6A Fig and S3 File; data available in S26 Data). Although the GO term cellular component represents a comprehensive annotation, it denotes the cellular compartments where processes such as signal transduction and membrane trafficking occur, essential for maintaining the cell homeostasis [66,67]. In *D. melanogaster*, we observed a strong effect of gene age on $\omega_a$ for proteins encoding chromosomal organisation, protein complex, stress response, signal transduction, and involved in the cell cycle (S6B Fig and S3 File; data available in S26 Data). Even though these functions cover a wide range of molecular processes, they are involved in DNA replication, genome stability, and immune and stress responses, which are critical functions for the coevolutionary arms race between hosts and parasites [47]. When looking at the nonadaptive rate of evolution, our analyses revealed a strong influence of gene age in $\omega_{na}$ within most functions analysed in both species, where young genes present higher rates of nonadaptive substitutions (S6 Fig and S3 File; data available in S26 Data). These results suggest that, when restricting the analysis to proteins involved in defence mechanisms, which are known to adapt faster [42,43,47,65], gene age still has an impact on the efficiency of selection acting upon a protein.

## Substitutions in young genes have larger effect sizes

Our second prediction under the adaptive walk model is that substitutions in young genes should have larger fitness effects than in older genes. To test this prediction, we used Grantham's physicochemical distances between amino acids [68] as a proxy for the fitness effects of amino acid mutations. We looked at the fixed differences separated by 1 mutational step between each pair of species (i.e., fixed differences in the ingroup when compared to the outgroup species) and reported the average Grantham's distances between residues within each age stratum. We observed that substitutions in young genes tend to occur between less biochemically similar residues (*Arabidopsis*: τ = 1, $p = 2.00 \times 10^{-7}$; *Drosophila*: τ = 0.788, $p = 3.628 \times 10^{-4}$; Fig 3 and S4 File; data available in S27 Data), suggesting that substitutions in these genes have larger physicochemical effects than in old ones. To further test whether these larger effects had a positive or negative impact on younger genes, we estimated the average Grantham's distance among adaptive ($\overline{G}_a$) and nonadaptive ($\overline{G}_{na}$) nonsynonymous substitutions (see Material and methods and S4 File; data available in S28 Data). We expect a positive correlation between $\overline{G}_a$ and $\overline{G}_{na}$ and gene age, i.e., the average Grantham's distance for adaptive and nonadaptive substitutions should increase towards the present. This is indeed what we find: We observed a positive correlation between $\overline{G}_a$ and gene age in both species, which is individually significant in *Drosophila* ($p = 0.006$) and significant across species when we combine probabilities ($p = 0.009$) (S7 Fig). We observed the same pattern in the analysis of $\overline{G}_{na}$ with a significant correlation in *Arabidopsis* ($p = 0.004$) and a significant correlation when combining the probabilities from both species ($p = 0.0007$) (S7 Fig).

## Discussion

We used a population genomic approach to disentangle the effects of positive and negative selection on the rate of nonsynonymous substitutions. Using complete genome data from 2 *Arabidopsis* and *Drosophila* species, we showed that the higher rate of nonsynonymous substitutions in younger genes results both from relaxed purifying selection (higher $\omega_{na}$) and a higher rate of adaptive substitutions (higher $\omega_a$) (Fig 1B). Although gene age is not the variable contributing most to the rate of molecular adaptation (S4 Table), its effect is consistent across comparisons and robust to multiple confounding factors (Fig 2 and Tables 1 and S2 and S3). Moreover, we observed that young genes undergo substitutions that are larger in terms of

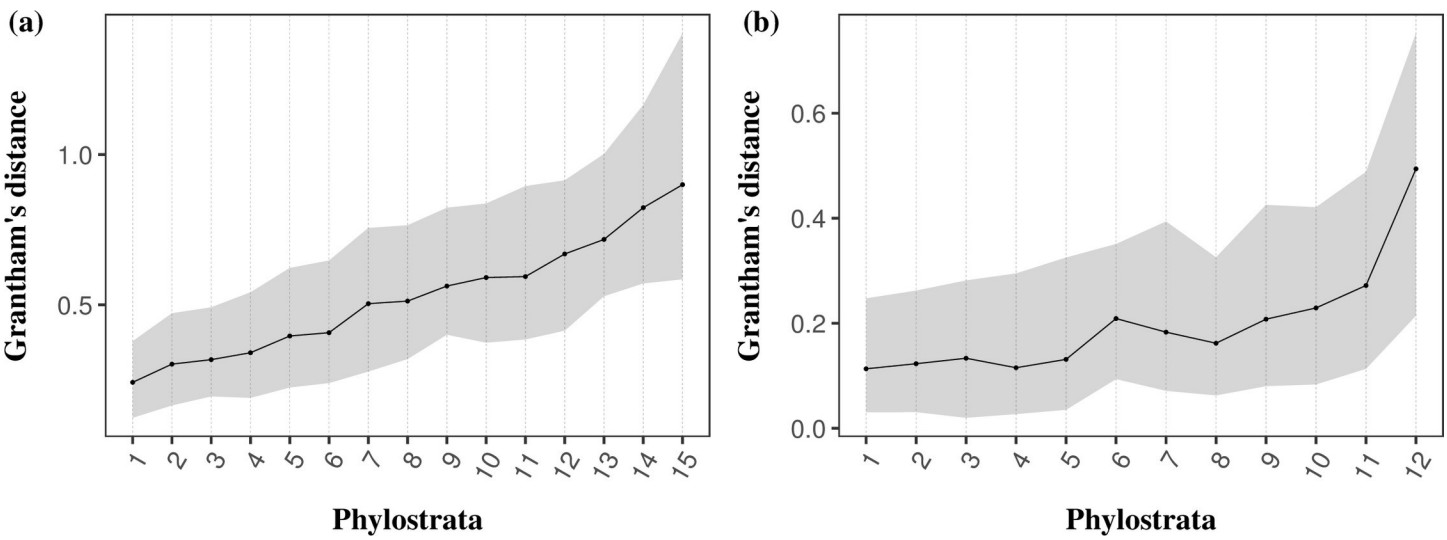

**Fig 3.** Relationship between gene age and Grantham's distance between amino acids for *A. thaliana* (a) and *D. melanogaster* (b). For each clade, the median value of Grantham's distance between residues is depicted with the black dot. The shaded area represents the physicochemical distances within the first and third quartile. The data (S27 Data) and code needed to generate this table can be found at https://gitlab.gwdg.de/molsysevol/supplementarydata_geneage and https://zenodo.org/record/6828430.

physicochemical properties than older genes. A question remains: What are the drivers of these effects?

## The magnitude effect of gene age on adaptive evolution is species-specific

Although we observed a strong impact of gene age on the molecular adaptive rate in both species pairs, the shape of the relation between these 2 variables differs. While the relationship between gene age and $\omega_a$ is monotonously increasing in *Arabidopsis*, it has several peaks in *Drosophila* (Fig 1B). This pattern is particularly evident if we discard the 2 youngest clades. In *Drosophila*, the correlation becomes much weaker and nonsignificant for $\omega_a$ ($\omega$: $\tau = 0.600$, $p = 0.016$; $\omega_{na}$: $\tau = 0.556$, $p = 0.025$; $\omega_a$: $\tau = 0.467$, $p = 0.060$), whereas in *Arabidopsis*, the effect of gene age persists ($\omega$: $\tau = 0.9487$, $p = 6.342 \times 10^{-6}$; $\omega_{na}$: $\tau = 0.872$, $p = 3.345 \times 10^{-5}$; $\omega_a$: $\tau = 0.692$, $p = 9.86 \times 10^{-4}$). Intriguingly, this multimodal distribution of $\omega_a$ observed in *Drosophila* resembles the pattern of gene emergence in this species [17]. The peak in the adaptive substitution rate observed for clades 6 and 7 (Fig 1B) coincided with the animal phyla's major radiation at the time of extensive periods of glacial cycles [69]. When looking at the functions coded by these proteins, we found that they are linked to a wide range of vital cellular and biological processes, such as defence mechanisms and cell differentiation (S8 Fig and S3 File; data available in S25 Data). This pattern suggests that these genes might be experiencing higher molecular adaptive rates due to their role in such vital processes. However, for these genes to keep such high rates of adaptive substitutions until recent times (i.e., in the branch between *D. melanogaster* and *D. simulans*), epistatic interactions might be at play. Studies across taxa have proposed that functional epistasis is an important factor in the evolution of genes involved in defence mechanisms and adaptation to new environmental stresses [70–74]. We posit that such gene interactions keep these proteins further from their optimum throughout time due to the rugged shape of the fitness landscape, leading to the high molecular adaptive rates observed in the branch between *D. melanogaster* and *D. simulans*. To further test this hypothesis, we used the degree of protein–protein interactions (PPIs) as a proxy for epistatic interactions and analysed its relationship with gene age. We observed that genes in clade 7 have a slightly higher

degree of PPI than other strata (S9 Fig and S5 File; data available in S29 Data), suggesting that these genes might be experiencing relatively more epistatic interactions. These findings are consistent with epistasis influencing the evolution of these genes, potentially explaining their continued higher rates of molecular adaptation. In contrast, the burst of the emergence of new genes in *Arabidopsis* coincided with the plant-specific radiation right before the emergence of Brassicaceae [17,75]. This trend is consistent with our results from *A. thaliana*, where the bursts of $\omega_a$ in younger clades (after clades 11 and 12 in Fig 1B). These distinct patterns observed between species suggest that the role of a gene's age in molecular adaptation is complex, as also evidenced by the lack of a significant correlation with $\omega_a$ previously reported in humans [22]. The authors proposed that this result may be a consequence of the generally low molecular adaptive rates observed in primates [22,48].

Despite these species-specific trends, our analyses revealed a strong correlation between $\omega_a$ and gene age extending through hundreds of millions of years (Figs 1 and 2 and S2 and S3 Tables). These findings suggest a consistent effect of a gene's age on the rate of molecular adaptation across taxa.

## An adaptive walk model of gene evolution

Our study highlighted that, after their emergence, young genes evolve through relaxed selection, as first proposed by Ohno [76], but also by acquiring beneficial mutations, as described in the "adaptive-conflict" model [37,77]. Ohno's idea of evolution was "non-Darwinian" in its nature, as he believed that "natural selection merely modified while redundancy created" [76]. He proposed that new genes evolve by accumulating "forbidden" mutations, where they are only preserved if the development of a formerly nonexistent function occurs, a process known as neofunctionalisation. In this scenario, natural selection only acts at the stage of acquiring a new function. Further extensions of this theory suggested that the preservation of a new gene can also occur through subfunctionalisation, where the accumulation of deleterious mutations leads to a complementary loss of function in both copies of the gene [78,79].

In contrast, the "adaptive-conflict" model assumes that the ancestral gene could carry more than 2 pleiotropically constrained functions [37,77]. Once the duplication event occurs, each copy then becomes specialised in one of the ancestral functions. In this case, the ancestral gene's split proceeded through positive Darwinian selection [37,77]. These theories are based on the evolution of gene duplicates and agree with the idea of evolution as a "tinkerer" proposed by Jacob [80], where evolution adjusts the already existing elements. In *de novo* evolution, however, new genes emerge by acquiring new functions from the noncoding fragments of the genome [17,81,82]. This process is thought to proceed through a stochastic phase followed by the successive accumulation of beneficial mutations, ultimately leading to a new function with a species-specific selective advantage [83–86].

When looking at the fundamental ideas behind these theories, one can draw 1 prominent feature that portrays the evolution of new genes: Young genes are further away from their fitness optimum. Hence, we posit that these genes follow an adaptive walk model of gene evolution to reach their fitness peak [3,4,87]. As their full potential has yet to be met, more consecutive beneficial mutations are theoretically needed to reach their fitness optimum, leading to the higher molecular adaptive rates observed in these genes. In turn, older genes are closer to their optimal features and less robust to large effects' mutations, thus only accumulating mutations with small fitness effects. Such slightly advantageous mutations are more difficult to select for, leading to lower adaptive rates in these proteins. We further tested this hypothesis using Grantham's physicochemical distances [68] as a proxy for the fitness effect of substitutions. This analysis showed that substitutions in young genes tend to occur between

more dissimilar residues (Fig 3), suggesting that the evolution of young genes proceeds in larger steps compared to old ones. Moreover, we observed that adaptive substitutions have greater physicochemical effects (i.e., the average Grantham's distance of adaptive substitutions ($\overline{G}_a$) is correlated to gene age; S7 Fig). However, the strength of this signal differs between species. While we observed a strong positive correlation between $\overline{G}_a$ and gene age in *Drosophila*, this relationship is weaker in *Arabidopsis*. The opposite pattern is observed for the correlation between $\overline{G}_{na}$ and gene age, where this relationship is only significant in *Arabidopsis* (S7 Fig). These patterns suggest that, in *Arabidopsis*, there is a stronger influence of relaxed purifying selection when compared to positive selection. This weaker adaptive signal may be a consequence of their mating system. The transition from out-crossing to self-fertilization in *A. thaliana* exposed these plants to less efficient purifying selection due to their reduced effective population sizes [88,89], thus accumulating more deleterious mutations. This higher load of segregating deleterious mutations can interfere with nearby beneficial mutations, preventing their fixation and slowing down adaptation. Indeed, previous studies have shown that segregating recessive deleterious mutations can substantially decrease the fixation rate of advantageous mutations, leading to a weaker adaptive signal in some parts of the genome [90,91]. In contrast, the large effective population sizes of *Drosophila* species make them more efficient in removing deleterious mutations and prone to stronger positive selection [92–94]. These patterns could explain the observed differences in the strength of selection in young genes between species.

An alternative hypothesis to the adaptive walk theory is a static model of adaptation. This model is based on the level of constraint of a gene: As some genes are less constrained than others, they accumulate both advantageous and deleterious mutations, some of which may lead to the loss of a gene. Under this hypothesis, less constrained genes tend to be young because they are more likely to be lost. Additionally, as such genes accumulate a larger number of deleterious mutations, they also represent a bigger mutational target for adaptive mutations, which could also explain the overall correlation between $\omega_a$ and $\omega_{na}$. We can test this hypothesis by assessing whether the correlation between $\omega_a$ and gene age is independent of $\omega_{na}$. We controlled for the effect of $\omega_{na}$ by running Grapes on a subset of genes for which the correlation between $p_n/p_s$ and gene age was no longer significant. We found that gene age was still significantly correlated with $\omega_a$ whereas $\omega_{na}$ was not (S10 Fig; data available in S30 Data), thus suggesting that the rate of adaptive evolution is independent of the nonadaptive rate. While these results do not completely discard the possibility of a stationary view of adaptation, they agree with an adaptive walk model of gene evolution.

Overall, our findings support an adaptive walk model hypothesis of gene evolution over time, where protein-coding genes adapt following a pattern of diminishing returns in *Drosophila* and *Arabidopsis*. The increasing availability of population genomic data sets will allow the detailed characterization of the adaptive walk model of gene evolution in an increasingly diverse range of taxa.

## Material and methods

We assessed the role of gene age on adaptive evolution using the divergence and polymorphism data published in Moutinho and colleagues [39]. For *D. melanogaster*, the data included 10,318 protein-coding genes in 114 genomes for 1 chromosome arm of the 2 large autosomes (2L, 2R, 3L, and 3R) and 1 sex chromosome (X) pooled from an admixed sub-Saharan population from Phase 2 of the *Drosophila* Genomics Project [DPGP2] [95] and divergence estimates from *D. simulans* (S31 Data). For *A. thaliana*, we analysed 18,669 protein-coding genes in 110 genomes comprising polymorphism data from a Spanish population (1001 Genomes Project)

[96] and divergence out to *A. lyrata* (S32 Data). We used these data sets to infer the synonymous and nonsynonymous unfolded site frequency spectrum (SFS), and synonymous and nonsynonymous divergence from the rate of synonymous and nonsynonymous substitutions. Sites with a missing outgroup allele were considered as missing data. All estimates were obtained using the "bppPopStats" program from the Bio++ Program Suite (version 2.4.0) [97]. Gene age data were obtained from published data sets, wherein 9,004 *Drosophila* [28] and 17,732 *Arabidopsis* [33] genes were used. Analyses were performed by dividing genes into 12 and 15 phylostrata for *D. melanogaster* and *A. thaliana* (Fig 1), respectively, numbered from the oldest (stratum 1) to the most recent (strata 12 and 15 in *D. melanogaster* and *A. thaliana*, respectively). The most recent clades include orthologous genes present in each species and their respective outgroups. The analyses on the X-linked and autosomal genes in *D. melanogaster* were performed with 1,478 and 7,526 genes, respectively. We fitted models of the DFE across different age classes and gene categories to estimate the molecular rate of adaptation [48].

## Estimation of the adaptive and nonadaptive rate of nonsynonymous substitutions

The rates of adaptive nonsynonymous substitutions were estimated with the Grapes program [48], using the Gamma-Exponential DFE, as this model was previously shown to best fit the data [39]. As this method does not accommodate SNPs with missing information in some individuals, we reduced the original data set to $n = 110$ and $n = 105$ genomes for *D. melanogaster* and *A. thaliana*, respectively, by randomly down-sampling sites with at least n genotypes available and discarding positions where less than n genotypes were present. Estimates of substitution rates and their confidence intervals were obtained with a bootstrap analysis by sampling genes in each category, with replacement. We performed a total of 100 replicates, and the DFE model was fitted for each replicate with Grapes. Results for $\omega$, $\omega_{na}$, and $\omega_a$ were plotted using the R package "ggplot2" [98] by taking the mean value and the 95% confidence interval of the 100 bootstrap replicates performed for each category (see detailed R scripts in the supplementary files in the supplementary data online, https://gitlab.gwdg.de/molsysevol/supplementarydata_geneage and https://zenodo.org/record/6828430).

## Gene age versus protein length and gene expression

Data on protein length and gene expression were included in the data sets of Moutinho and colleagues [39] (S31 and S32 Data). Gene expression data were downloaded from the database Expression Atlas [99], taking 1 baseline experiment for each species [39]. Mean gene expression levels were then obtained by averaging across samples and tissues for each gene. To correct for these 2 factors, we divided our data set into 2 equally sized groups based on the factor we wished to control. Short proteins had a size up to 366 and 389 amino acids, and long proteins had a size up to 4,674 and 5,098 amino acids in *A. thaliana* and *D. melanogaster*, respectively. We further merged phylostrata containing a low number of genes. For *D. melanogaster*, we categorised gene age into 6 main clades by combining clades 3 to 4, 5 to 6, 7 to 10, and 11 to 12, keeping the others unchanged. In *A. thaliana*, we combined the 15 clades into 6 main groups by merging clades 5 to 8 and 9 to 15. For gene expression, we used a total of 17,126 and 6,247 genes for *A. thaliana* and *D. melanogaster*, respectively, being categorised as lowly and highly expressed. Genes were classified as lowly expressed if the mean expression levels were up to 10.3 and 6.8, and highly expressed genes were the ones with expression up to 6,632.8 and 4,237.0 in *A. thaliana* and *D. melanogaster*, respectively. For *D. melanogaster*, we categorised

gene age into 6 categories by combining clades 3 to 5, 6 to 9, and 10 to 12. In *A. thaliana*, we combined the data in 6 clades, merging clades 4 to 7, 8 to 11, and 12 to 15 (S6 File).

## Gene age versus protein structure

Since most young genes lack a defined three-dimensional structure [36], they do not have information on the residue's solvent accessibility. Hence, we used a deep learning approach, NetSurfP-2.0, which predicts the RSA of each residue from the amino acid sequence [100] by using as a training model the HH-suite sequence alignment tool for protein similarity searches [101]. To assess whether this approach provided reliable results, we compared the RSA estimates of NetSurfP-2.0 with those obtained from the PDB structures in our data set [39]. We found a good correlation between the 2 approaches for both species (Kendall's $\tau = 0.571$, $p < 2 \times 10^{-216}$; $\tau = 0.462$, $p < 2 \times 10^{-216}$, for *D. melanogaster* and *A. thaliana*, respectively). Using NetSurfP-2.0, RSA estimates were successfully obtained for a total of 4,238,686 (88% of the total codon sites) and 7,479,807 (99% of the total codon sites) amino acid residues for *D. melanogaster* and *A. thaliana*, respectively. To assess the impact of RSA at the gene level, we analysed the total number of genes in both species by making 2 categories of genes according to the average RSA value per gene. Genes with lower RSA had mean values between 0.127 and 0.389 in *Drosophila* and 0.217 and 0.386 in *Arabidopsis*. Genes with a higher RSA had mean values between 0.390 and 0.894 in *Drosophila* and 0.387 and 0.898 in *Arabidopsis*. The phylostrata groups were defined by combining clades 7 to 8 in *D. melanogaster* and 8 to 11 and 12 to 15 in *A. thaliana* (S6 File).

For the analysis of the residue intrinsic disorder, we used the estimates included in the data sets of Moutinho and colleagues [39]. These data were obtained with the software DisEMBL [102], where intrinsic disorder was estimated per site and classified according to the degree of "hot loops," i.e., highly mobile loops. This analysis was performed for 17,732 and 7,410 genes for *A. thaliana* and *D. melanogaster*, respectively. Genes were combined into 2 categories according to the mean value of their residue's intrinsic disorder. Genes with a low level of intrinsic disorder had values between 0.029 and 0.080 in *Drosophila* and among 0.041 and 0.084 in *Arabidopsis*. Genes with a higher degree of intrinsic disorder had values between 0.081 and 0.554 in *Drosophila* and among 0.085 to 0.551 in *Arabidopsis*. In *D. melanogaster*, all the 12 phylostrata could be used. In *A. thaliana*, the 15 strata were combined in 12 categories by merging clades 9 to 10, 11 to 12, and 13 to 14 (S6 File).

## Correcting for BLAST E-values

We analysed the robustness of the gene age's effect by correcting the variation in the Expect (E) value estimates in BLAST's searches between our focus species and their respective outgroups. By reducing the variation in E-values estimates, we could correct for potential failures in BLAST's homology searches. To do so, we limited the analysis to genes with particularly low E-values that are more likely to be identified in all age strata: 12,472 genes with an E-value lower than $1 \times 10^{-150}$ for *A. thaliana* and 7,104 genes with an E-value lower than $1 \times 10^{-100}$ for *D. melanogaster* (S2 File). For *A. thaliana*, analyses were carried out by combining clades 8 to 13, with no genes left in clades 14 and 15. For *D. melanogaster*, analyses were performed with the 12 strata. By using these genes, we found no correlation between E-value and age.

## Gene age versus protein function

GO terms were obtained from the "dmelanogaster_gene_ensembl" and the "athaliana_eg_gene" tables in the Ensembl database (version 103), through the R package "biomaRt" [103]. A total of 7,253 (approximately 70% of the genes) and 15,604 (approximately 80% of the genes)

genes were mapped in *D. melanogaster* and *A. thaliana*, respectively. To check whether the effect of gene age prevailed across functional protein classes, we analysed the GO terms with the highest number of young genes mapped: more than 50 genes present in clades 11 and 12 in *D. melanogaster* and more than 30 genes present in clades 12 to 15 in *A. thaliana*. This filtering step resulted in 6,637 genes across 23 GO categories in *D. melanogaster* (S31 Data) and 15,410 genes across 10 GO categories in *A. thaliana* (S32 Data). To analyse the effect of gene age, we compared 3 age classes. In *D. melanogaster*, the first age category spanned over phylostrata 1 to 3, the second category covered clades 4 to 7, and the third one included clades 8 to 12. In *A. thaliana*, the first category comprised genes belonging to clades 1 to 6, the second category spanned over clades 7 to 11, and the third one included the phylostrata between clades 12 and 15 (Fig 1A).

## Gene age versus protein–protein interactions (PPIs)

We obtained PPI data for *D. melanogaster* from the STRING database [104], which includes both physical and functional interactions (https://string-db.org/). This database included 13,046 proteins with annotated interactions, which were used to analyse the distribution of protein networks across phylostrata.

## The fitness effects of amino acid substitutions

We used Grantham's physicochemical distances between amino acids [68] as a proxy for the fitness effects of amino acid substitutions and performed 2 analyses (see detailed R script in S4 File). In the first one, we calculated the average Grantham's distance between amino acid substitutions for each phylostrata. In the second analysis, we estimated the average Grantham's distance for adaptive ($\overline{G}_a$) and nonadaptive ($\overline{G}_{na}$) nonsynonymous substitutions within each age stratum. This analysis was performed by running Grapes across amino acid pairs separated by a single mutational step within categories of gene age. To run Grapes, we first estimated the synonymous and nonsynonymous SFS using the same approach as in Bergman and Eyre-Walker [105]. This method compares the nonsynonymous SFS for a particular amino acid pair, for example, alanine and aspartic acid, which are separated by a $A <> C$ mutation, with the synonymous SFS of 4-fold degenerate sites separated only by $A <> C$ mutations ($SFS_{4F}$ $_{(A<>C)}$). For amino acid pairs separated by more than 1 mutation type, we estimated the weighted average of the synonymous SFS of 4-fold sites for the different types of mutations, weighting by the frequency of the respective codons. For example, serine and cysteine are separated by $A <> T$ and $C <> G$ mutations. The synonymous SFS for this amino acid pair was estimated as $SFS_{4F}(weighted) = [(f_{AGT} + f_{TGT} + f_{AGC} + f_{TGC}) * SFS_{4F(A<>T)} + (f_{TCT} + f_{TGT} + f_{TCC} + f_{TGC}) * SFS_{4F(C<>G)}] / (f_{AGT} + f_{TGT} + f_{AGC} + f_{TGC} + f_{TGT} + f_{TGC}$, where $f$ corresponds to the frequency of each codon (e.g., $f_{AGT}$ is the frequency of the codon AGT). It was not possible to estimate the rates of adaptive and nonadaptive substitution for each pair of amino acids for each phylostratum because we have insufficient data. So, we combined amino acid pairs into 10 categories based on their Grantham's distance, and we combined clades 5 to 6, 8 to 11, and 12 to 15 in *Arabidopsis* and clades 3 to 4, 5 to 7, and 8 to 10 in *Drosophila*. We then ran Grapes to estimate $\omega_a$ and $\omega_{na}$ for the 10 categories of Grantham's physicochemical distances for each combined age stratum. We further calculated the average Grantham's distance for adaptive ($\overline{G}_a$) and nonadaptive ($\overline{G}_{na}$) nonsynonymous substitutions using the values of $\omega_a$ and $\omega_{na}$ estimated with Grapes and applying the following equations: $\overline{G}_a = \sum_i \omega_{ai} \overline{g}_l N_i / \sum_i N_i$ and $\overline{G}_{na} = \sum_i \omega_{nai} \overline{g}_l N_i / \sum_i N_i$, where the sum is across the 10 categories of Grantham's distances within each age class, $\omega_{ai}$ and $\omega_{nai}$ are the rates of adaptive and nonadaptive substitution in the

$i$th category for each age class, $\overline{g_i}$ is the average Grantham's distance among the amino acid pairs within each Grantham's distance category, and $N_i$ is the number of nonsynonymous sites in the $i$th category in each age class. Higher values of $\overline{G}_a$ (the same for $\overline{G}_{na}$ with nonadaptive substitutions) mean that there is a higher proportion of adaptive substitutions across Grantham's distance categories.

## Assessing the correlation between $\omega_a$ and $\omega_{na}$

To test whether the impact of gene age on $\omega_a$ was independent of $\omega_{na}$, we ran Grapes for a subset of genes for which the correlation between $\omega_{na}$ and gene age was no longer significant (see detailed R script in S7 File). We controlled for the variation in $\omega_{na}$ by discarding genes according to their $p_n/p_s$ values. To ensure a sufficient number of genes, we kept the genes that were within 0.02 and 0.05 standard deviation from the modal value of the $p_n/p_s$ distribution in *Arabidopsis* (mode: 0.05) and *Drosophila* (mode: 0.01), respectively. The subset of data included 8,652 and 2,570 genes for *Arabidopsis* and *Drosophila*, respectively.

## Statistical analyses

Assessing the effect of gene age within each protein functional class was performed by comparing rate estimates between all pairs of age categories. A total of 100 bootstrap replicates were generated and $\omega_a$ and $\omega_{na}$ were estimated for each resampling, allowing us to compute the rate differences between categories. A one-tailed $p$-value can be obtained using the formula P = (k + 1)/ (N + 1), where N = 100 is the number of bootstrap replicates and k is the number of times the computed difference was greater (resp. lower) than 0. Here, we used a two-tailed version of this test, computing the $p$-value as P = [2 * min (k$^-$, k$^+$) + 1] / (N + 1), where k$^-$ is the number of times the difference was negative, and k$^+$ is the number of times the difference was positive. *P*-values for all pairwise comparisons were corrected for multiple testing using the FDR method [106] as implemented in R [107] (see detailed R script in S3 File). For the analysis with PPI and gene age, statistical significance was assessed using nonparametric post hoc tests, as implemented in the "Kruskal" method of the R package "agricolae" using the FDR method to correct for multiple testing [108] (see detailed R script in S5 File). For the rest of the analyses, statistical significance was assessed with Kendall's correlation tests using the mean value of the 100 bootstrap replicates for each category (see detailed script in S6 File). To estimate the combined $p$-value for each cofactor, we used the weighted-Z method using the R package "metap" [109]. We estimated the weight of each $p$-value using a linear modelling approach with $\omega_a$ and $\omega_{na}$ as response variables and gene age and potential cofactors as explanatory variables and inferred the reciprocal of the squared standard error of the residuals in each model (see detailed R scripts in S8 File). To determine whether the chromosome impacted gene age's effect on estimates of $\omega_a$ and $\omega_{na}$, we performed an analysis of covariance (ANCOVA) by comparing a model M1 that included the impact of the chromosome, age, and their interaction, with a model M0 that included age only (see detailed R script in S1 File). Normality, homoscedasticity, and independence of the error terms of the model were assessed with the package "lmtest" [110] in R.

 Finally, to further assess the effect of residual correlations between gene age and each cofactor, we fitted linear models with the rates of protein evolution (i.e., $\omega$, $\omega_{na}$ and $\omega_a$) as response variables and the average value of the cofactor for each age class as an explanatory variable (nested within the cofactor category, low or high), together with gene age, species (*Drosophila* or *Arabidopsis*), cofactor category, and their interaction. A model selection was conducted

using the R function "step" and the significance of the model coefficients was assessed for the selected models (see detailed R script in S9 File).

## MK regression

The MK regression analysis was performed per nucleotide site, where we analysed the combined effect of RSA, intrinsic protein disorder, protein length, gene expression, protein function, Grantham's distances, and the sex chromosome in *Drosophila*. As this method does not automatically handle categorical variables, we created a binary variable for protein function based on the gene's presence (1) or absence (0) in stress-related proteins (i.e., response to stress, response to external stimulus and signal transduction). We used the same rationale for the X chromosome effect in *Drosophila*. We performed the MK regression analysis by comparing polymorphic and divergence sites between nonsynonymous and 4-fold degenerated sites in both species. As this method does not account for the effects of weak selection [57], we analysed only sites with an allele frequency above 50%. To assess the relative contribution of each variable to the regression model, we compared the likelihood pseudo-$R^2$ following the Cox and Snell method [59]. These were estimated by comparing the log-likelihood of the full model (all factors included) with that from each reduced model (each factor removed) using the following equation:

$$\text{R2} = 1 - \left( \exp\left( -\frac{2}{N} * (ln_{full} - ln_{reduced}) \right) \right)$$

where $N$ represents the number of sites analysed with the MK regression, $ln_{full}$ represents the log-likelihood of the model with all factors included, and $ln_{reduced}$ represents the log-likelihood of the model excluding each factor (e.g., to assess the contribution of the gene age variable, the full model was compared to a model including all variables except gene age).

## Supporting information

**S1 Table. Kendall's correlation coefficients for the relationship between the mean value of each cofactor in each age class and ω, ω_na, and ω_a for each "high" and "low" group.** The Kendall's correlation coefficients for the relationship between the cofactor and gene age in each "high" and "low" group are presented in the column "co-factor~Age". The data (S14–S18 Data tables) and code needed to generate this table can be found at https://gitlab.gwdg.de/molsysevol/supplementarydata_geneage and https://zenodo.org/record/6828430. (XLSX)

**S2 Table. Linear models' estimates when using the rates of protein evolution (ω, ω_na, and ω_a) as response variables and the mean value of the cofactor per category as a putative explanatory variable, together with gene age, the species (*Arabidopsis* or *Drosophila*), cofactor-category, and their interaction with gene age.** The data (S14–S17 Data tables) and code needed to generate this table can be found at https://gitlab.gwdg.de/molsysevol/supplementarydata_geneage and https://zenodo.org/record/6828430. (XLSX)

**S3 Table. MK regression estimates, z-scores, and respective *p*-values. The data (S19–S22 Data tables) needed to generate this table can be found at https://gitlab.gwdg.de/molsysevol/supplementarydata_geneage and https://zenodo.org/record/6828430.** (XLSX)

**S4 Table. Partial $R^2$ estimates for each factor analysed in the MK regression analysis.** (XLSX)

**S1 Fig. Relationship between gene age and gene length (a) and gene expression (b) for *A. thaliana* (top) and *D. melanogaster* (bottom).** This analysis was performed by categorizing gene age according to the clades defined in Fig 1A. For each clade, the median value of gene length and gene expression is depicted with the black dot. The shaded area represents the values of gene length and mean expression levels within the first and third quartile. The data (S2 and S3 Data tables) and code needed to generate this figure can be found at https://gitlab. gwdg.de/molsysevol/supplementarydata_geneage and https://zenodo.org/record/6828430. (PDF)

**S2 Fig. Relationship between gene age and RSA (a) and protein intrinsic disorder (b) for *A. thaliana* (top) and *D. melanogaster* (bottom).** Legend as in S1 Fig. The data (S4 and S5 Data tables) and code needed to generate this figure can be found at https://gitlab.gwdg.de/ molsysevol/supplementarydata_geneage and https://zenodo.org/record/6828430. (PDF)

**S3 Fig. Relationship between gene age and E-values before (a) and after (b) the E value correction for *A. thaliana* (top) and *D. melanogaster* (bottom).** Each black dot represents a gene and median E value for each clade is represented with the black line in the boxplot. The data (S23 Data) and code needed to generate this figure can be found at https://gitlab.gwdg.de/ molsysevol/supplementarydata_geneage and https://zenodo.org/record/6828430. (PDF)

**S4 Fig. Estimates of $\omega$, $\omega_{na}$, and $\omega_a$ plotted as a function of gene age by correcting for the E-value on BLAST's searches in *A. thaliana* (top) and *D. melanogaster* (bottom).** Mean values of $\omega$, $\omega_{na}$, and $\omega_a$ for each category are represented with the black points. Error bars denote for the 95% confidence interval for each category, computed over 100 bootstrap replicates. The data (S24 Data) and code needed to generate this figure can be found at https://gitlab.gwdg.de/ molsysevol/supplementarydata_geneage and https://zenodo.org/record/6828430. (PDF)

**S5 Fig. The number of young genes for the respective protein function in (a) *D. melanogaster* and (b) *A. thaliana*.** The data (S25 Data) and code needed to generate this figure can be found at https://gitlab.gwdg.de/molsysevol/supplementarydata_geneage and https://zenodo. org/record/6828430. (PDF)

**S6 Fig. Estimates of $\omega$, $\omega_{na}$, and $\omega_a$ plotted as a function of protein function and gene age in (a) *A. thaliana* and (b) *D. melanogaster*.** Categories are ordered according to the values of $\omega_a$. Gene age categories are ordered from old (1) to young (3). Mean values of $\omega$, $\omega_{na}$, and $\omega_a$ for each class are represented with the black points. Error bars denote the 95% confidence interval for each category, computed over 100 bootstrap replicates. The data (S26 Data) and code needed to generate this figure can be found at https://gitlab.gwdg.de/molsysevol/ supplementarydata_geneage and https://zenodo.org/record/6828430. (PDF)

**S7 Fig. Relationship between the proportion of adaptive ($\overline{G_a}$) and nonadaptive ($\overline{G_{na}}$) substitutions and gene age.** Each point represents the weighted average for each age category. A linear model was fitted between gene age and Grantham's distances values and is represented with the blue line. Statistical significance was assessed with a Pearson's correlation test and the respective correlation coefficient (R) and $p$-values (p) are shown in each plot. The data (S28 Data) and code needed to generate this figure can be found at https://gitlab.gwdg.de/

molsysevol/supplementarydata_geneage and https://zenodo.org/record/6828430.
(PDF)

**S8 Fig. Frequency of genes belonging to clades 6–7 for the respective protein function in *D. melanogaster*.** The frequency of genes for each functional category was estimated by dividing the number of genes present in these clades by the total number of genes annotated with the respective function. The data (S25 Data) and code needed to generate this figure can be found at https://gitlab.gwdg.de/molsysevol/supplementarydata_geneage and https://zenodo.org/record/6828430.
(PDF)

**S9 Fig. Distribution of the degree of PPI values in *D. melanogaster*.** The statistical group for each clade is represented. The black line represents the median value of PPI for each clade and black dots denote the outliers of the distribution. The y-axis is scaled with a square root function. The data (S29 Data) and code needed to generate this figure can be found at https://gitlab.gwdg.de/molsysevol/supplementarydata_geneage and https://zenodo.org/record/6828430.
(PDF)

**S10 Fig. Estimates of $\omega$, $\omega_{na}$, and $\omega_a$ plotted as a function of gene age by correcting for the variation in $p_n/p_s$ in *A. thaliana* (top) and *D. melanogaster* (bottom).** The Kendall's correlation coefficients are shown with the respective significance ($^*P < 0.05$; $^{**}P < 0.01$; $^{***}P < 0.001$; "." $0.05 \leq P < 0.10$). Legend as in S4 Fig. The data (S30 Data) and code needed to generate this figure can be found at https://gitlab.gwdg.de/molsysevol/supplementarydata_geneage and https://zenodo.org/record/6828430.
(PDF)

**S1 Data. Gene age analysis (Fig 1B).**
(CSV)

**S2 Data. Correlation between gene age and protein length (S1A Fig).**
(CSV)

**S3 Data. Correlation between gene age and gene expression (S1B Fig).**
(CSV)

**S4 Data. Correlation between gene age and RSA (S2A Fig).**
(CSV)

**S5 Data. Correlation between gene age and protein disorder (S2B Fig).**
(CSV)

**S6 Data. Correlation between gene age and $\omega$, $\omega_{na}$, and $\omega_a$ while controlling for protein length (Fig 2A).**
(CSV)

**S7 Data. Correlation between gene age and $\omega$, $\omega_{na}$, and $\omega_a$ while controlling for gene expression (Fig 2B).**
(CSV)

**S8 Data. Correlation between gene age and $\omega$, $\omega_{na}$, and $\omega_a$ while controlling for RSA (Fig 2C).**
(CSV)

**S9 Data. Correlation between gene age and ω, $\omega_{na}$, and $\omega_a$ while controlling for protein disorder (Fig 2D).**
(CSV)

**S10 Data. Statistical results of the correlation between gene age and ω, $\omega_{na}$, and $\omega_a$ within each category of protein length.**
(CSV)

**S11 Data. Statistical results of the correlation between gene age and ω, $\omega_{na}$, and $\omega_a$ within each category of gene expression.**
(CSV)

**S12 Data. Statistical results of the correlation between gene age and ω, $\omega_{na}$, and $\omega_a$ within each category of RSA.**
(CSV)

**S13 Data. Statistical results of the correlation between gene age and ω, $\omega_{na}$, and $\omega_a$ within each category of protein disorder.**
(CSV)

**S14 Data. Correlation between the mean value of RSA within each category and ω, $\omega_{na}$, and $\omega_a$.**
(CSV)

**S15 Data. Correlation between the mean value of protein disorder within each category and ω, $\omega_{na}$, and $\omega_a$.**
(CSV)

**S16 Data. Correlation between the mean value of protein length within each category and ω, $\omega_{na}$, and $\omega_a$.**
(CSV)

**S17 Data. Correlation between the mean value of gene expression within each category and ω, $\omega_{na}$, and $\omega_a$.**
(CSV)

**S18 Data. Correlation between the mean value of each cofactor within each category and gene age.**
(CSV)

**S19 Data. Input data for 4-fold degenerate sites to run the MKRegression in *Drosophila*.**
(GZ)

**S20 Data. Input data for 0-fold degenerate sites to run the MKRegression in *Drosophila*.**
(GZ)

**S21 Data. Input data for 4-fold degenerate sites to run the MKRegression in *Arabidopsis*.**
(GZ)

**S22 Data. Input data for 0-fold degenerate sites to run the MKRegression in *Arabidopsis*.**
(GZ)

**S23 Data. Correlation between E-values and gene age before and after the correction (S3 Fig).**
(CSV)

**S24 Data. Correlation between gene age and ω, $\omega_{na}$, and $\omega_a$ after controlling for the variation in E-values (S4 Fig).**
(CSV)

**S25 Data. Correlation between gene age and protein function (S5 and S8 Figs).**
(CSV)

**S26 Data. Correlation between gene age and ω, $\omega_{na}$, and $\omega_a$ while controlling for protein function (S6 Fig).**
(CSV)

**S27 Data. Correlation between gene age and Grantham's distance (Fig 3).**
(CSV)

**S28 Data. Correlation between gene age and $\overline{G_a}$ and $\overline{G_{na}}$ (S7 Fig).**
(CSV)

**S29 Data. Correlation between gene age and protein–protein interactions (S9 Fig).**
(GZ)

**S30 Data. Correlation between gene age and ω, $\omega_{na}$, and $\omega_a$ after controlling for $p_n/p_s$ (S10 Fig).**
(CSV)

**S31 Data. All data per gene for *Drosophila melanogaster*.**
(GZ)

**S32 Data. All data per gene for *Arabidopsis thaliana*.**
(GZ)

**S1 File. Detailed R script of the gene age analysis.**
(PDF)

**S2 File. Detailed R script of the correlation between gene age and ω, $\omega_{na}$, and $\omega_a$ while controlling for the variation in E-values.**
(PDF)

**S3 File. Detailed R script of the correlation between gene age and ω, $\omega_{na}$, and $\omega_a$ while controlling for protein function.**
(PDF)

**S4 File. Detailed R script of the correlation between gene age and Grantham's distances.**
(PDF)

**S5 File. Detailed R script of the correlation between gene age and protein–protein interactions (PPI).**
(PDF)

**S6 File. Detailed R script of the correlation between gene age and ω, $\omega_{na}$, and $\omega_a$ while controlling for the protein length, gene expression, RSA, and protein intrinsic disorder.**
(PDF)

**S7 File. Detailed R script of the correlation between gene age and ω, $\omega_{na}$, and $\omega_a$ while controlling for the variation in $p_n/p_s$.**
(PDF)

**S8 File. Detailed R script of the analysis of the combined probabilities for each of the cofactors within and across species using the weighted Z-method.**
(PDF)

**S9 File. Detailed R script of the extra analyses performed to correct for intracategory variation of the cofactors.**
(PDF)

## Acknowledgments

The authors thank Diethard Tautz, Tal Dagan, and Chaitanya Gokhale for fruitful discussions.

## Author Contributions

**Conceptualization:** Ana Filipa Moutinho, Adam Eyre-Walker, Julien Y. Dutheil.

**Data curation:** Ana Filipa Moutinho.

**Formal analysis:** Ana Filipa Moutinho.

**Investigation:** Ana Filipa Moutinho, Adam Eyre-Walker, Julien Y. Dutheil.

**Methodology:** Ana Filipa Moutinho, Adam Eyre-Walker, Julien Y. Dutheil.

**Project administration:** Julien Y. Dutheil.

**Resources:** Julien Y. Dutheil.

**Software:** Ana Filipa Moutinho.

**Supervision:** Julien Y. Dutheil.

**Validation:** Ana Filipa Moutinho.

**Visualization:** Ana Filipa Moutinho.

**Writing – original draft:** Ana Filipa Moutinho, Julien Y. Dutheil.

**Writing – review & editing:** Ana Filipa Moutinho, Adam Eyre-Walker, Julien Y. Dutheil.

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
