## [Editor Report · Decision Letter 0]

10 Nov 2021

Dear Dr Moutinho, 

Thank you for submitting your manuscript entitled "Testing the adaptive walk model of gene evolution" for consideration as a Research Article by PLOS Biology.

Your manuscript has now been evaluated by the PLOS Biology editorial staff, as well as by an academic editor with relevant expertise, and I'm writing to let you know that we would like to send your submission out for external peer review.

Once your full submission is complete, your paper will undergo a series of checks in preparation for peer review. Once your manuscript has passed the checks it will be sent out for review. 

If your manuscript has been previously reviewed at another journal, PLOS Biology is willing to work with those reviews in order to avoid re-starting the process. Submission of the previous reviews is entirely optional and our ability to use them effectively will depend on the willingness of the previous journal to confirm the content of the reports and share the reviewer identities. Please note that we reserve the right to invite additional reviewers if we consider that additional/independent reviewers are needed, although we aim to avoid this as far as possible. In our experience, working with previous reviews does save time. 

If you would like to send your previous reviewer reports to us, please specify this in the cover letter, mentioning the name of the previous journal and the manuscript ID the study was given, and include a point-by-point response to reviewers that details how you have or plan to address the reviewers' concerns. Please contact me at the email that can be found below my signature if you have questions. 

Please re-submit your manuscript within two working days, i.e. by Nov 12 2021 11:59PM.

Kind regards,

Roli Roberts

Roland Roberts

Senior Editor

PLOS Biology

rroberts@plos.org

---

## [Decision Letter · Decision Letter 1]

20 Jan 2022

Dear Dr Moutinho,

Thank you for submitting your manuscript "Testing the adaptive walk model of gene evolution" for consideration as a Research Article at PLOS Biology. Your manuscript has been evaluated by the PLOS Biology editors, an Academic Editor with relevant expertise, and by three independent reviewers.

You'll see that while the reviewers are broadly positive about your study, they each raise a number of concerns that must be addressed before further consideration. These include several potential confounders that may not be fully controlled for, requests for further methodological clarification, concerns about your statistical treatment, some potential alternative explanations, and requests for more open-minded interpretation.

In light of the reviews (below), we will not be able to accept the current version of the manuscript, but we would welcome re-submission of a much-revised version that takes into account the reviewers' comments. We cannot make any decision about publication until we have seen the revised manuscript and your response to the reviewers' comments. Your revised manuscript is also likely to be sent for further evaluation by the reviewers.

We expect to receive your revised manuscript within 3 months. 

**IMPORTANT - SUBMITTING YOUR REVISION**

*Re-submission Checklist*

*Published Peer Review*

*PLOS Data Policy*

*Blot and Gel Data Policy*

Sincerely,

Roli Roberts

Roland Roberts

Senior Editor

PLOS Biology

rroberts@plos.org

REVIEWERS' COMMENTS:

Reviewer #1:

The manuscript by Moutinho et al. seeks to examine molecular adaptation and adaptive walks in natural populations of Drosophila and Arabidopsis. The adaptive walk model predicts that the initial steps of adaptation will consist primarily of many mutations of larger effects. Then, as the population gets closer to the fitness optimum, more small-effect mutations will predominate. Testing this prediction has been challenging for a number of reasons, some of which are related to the myriad of genomic confounders of genomic studies in natural populations. In this study, the authors hypothesize that newly arisen genes will be in the earlier stages of adaptive walks, experiencing more adaptive mutations and larger effect mutations while older genes will have smaller effect mutations. Indeed, by analyzing patterns of polymorphism and divergence in multiple species, the authors find support for this model. Specifically, they find that younger genes tend to have a higher rate of adaptive evolution than older genes. Further, substitutions in younger genes appear to result in changes that are less chemically similar to substitutions seen in older genes.

Overall, I found this paper to be a very creative and elegant study of an important fundamental question in evolutionary biology. The methods appear generally robust and the manuscript is clearly written. However, I have a number of concerns and suggestions on how to improve the manuscript.

Major comments:

1. Figure 2 and related analyses of confounders: I appreciate that the authors have attempted control for many of the confounders of other factors like gene expression level, protein length, and the amount of intrinsic disorder among proteins that can affect the rate of evolution and rate of adaptation. Indeed, the correlation between rate of adaptation and gene age seems to persist, even when only analyzing genes that are matched on the confounder. However, the analyses seemed to only control for a single confounder at a time. Do associations between the rate of adaptation and the other confounders persist when controlling for a single confounder. Put another way, does the association between rate of adaptation and protein length persist when controlling for expression level, for example? I wonder whether the main effect of gene age on evolutionary rate would still persist when somehow controlling for multiple confounders simultaneously? I realize that the authors indicate they cannot stratify into additional groups because they will not have enough genes in each group for a meaningful comparison. However, might some of the confounders also be correlated with each other? If so, might it possible to control for the joint effects of the confounder? 

2. Related to point 1 above and controlling for confounders—In figure 2 the authors report diving the genes into 2 equal groups based on the confounder and then testing for the main effect within each group. I'm not sure whether this is sufficient to adequately remove the effects of the confounder. For example, in Figure 2b, do genes in the "high expression" group still show a correlation between gene expression and rates of evolution? I worry that there will still be some variability in gene expression within each of the two large groups that could account for some of the association between the gene age and the rate of evolution. My comment holds for the other confounders as well.

3. Lines 212-241: The authors correctly are concerned that young and old genes may have different biological functions. If the young and old genes have different biological function, they may have different distributions of fitness effects and could have different rates of adaptation because of the differences in gene function, rather than the genes only differing in terms of where they are in adaptative walks. The authors attempt to control for this confounder using GO analyses. However, I'm not sure their control is adequate. Might it be possible to match the sets of "younger" and "older" genes based on having a similar set of GO terms and then testing whether the rates of adaptation differ between the younger and older genes? This matching on GO terms might better ensure that genes being compared have similar functional properties.

4. Lines 493-508: I like the analysis of amino acid exchangeability and gene age. However, I had a really hard time understanding how "G_a" and "G_na" were estimated in these analyses. I found the explanation in the Methods section to be rather vague and unclear, For example, "f_AGT" wasn't defined. Related, how well does this method using the SFS to estimate the rate of adaptation actually work for estimating G_a? Given the importance of these results for the overall conclusions of the paper, I think more description is required as well as some way of showing that the methods work. I tried looking at Bergman and Eyre-Walker [106], but that didn't help clarify things as much as needed.

Minor comments:

1. Line 76: Also, experimental studies are limited to only certain organisms.

2. Line 204: The use of the word "prevailed" here seems a little awkward. Maybe say "remained" or "persisted" instead.

3. It would be informative to include the P-values somehow on each plot in Figure 2.

4. Figure S7: P_a and P_na are listed in the caption but G_a and G_na are listed in the figure labels. Are these the same thing? Please clarify. I'm also confused by the point of this figure more generally. Please explain more.

Reviewer #2:

In this manuscript the authors test Fisher's geometric model and the idea of the adaptive walk where populations further away from the fitness optimum are more likely to fix beneficial mutations of larger effect first and of smaller effects later. They do so by testing this hypothesis in old vs new genes in Arabidopsis and Drosophila. Overall, I find the idea interesting and definitely worth testing. However, I'm not convinced of the results. Here are some specific and detailed comments:

Major Comments:

- I'm not entirely convinced of the results, which seem a bit over-sold throughout the manuscript without the explicit listing of caveats. 

a) For instance, from reading the abstract it appears that you have really estimated the selection coefficients of the adaptive substitutions; however you have used only a proxy for that.

b) Table S1 does show a positive correlation with gene age but there's a much stronger correlation with other factors like gene length and RSA. 

c) I'm also not convinced of the positive correlations between w_a and gene age, especially after you account for gene length and disorder (Figure 2a Arabidopsis; Figure 2d both). They appear extremely weak, unfortunately.

I would urge the authors to tone down the interpretation of their observations and discuss the caveats more.

-It would be a good idea to more thoroughly describe the idea behind using Grantham's distance in the Results section. It's not exactly a standard test and is not used very widely.

-In this particular study, I'm not really sure that using Grantham's distance is the best way to show that fitness effects of substitutions are large for young genes. If young genes are more likely to be disordered, they might also be less likely to have strong fitness effects when the physicochemical distances are larger. So this appears to be a confounding factor here. Have you looked at the correlations between Grantham's distance and gene age while accounting for the confounding factors? If I missed it, I apologize!

Minor Comments:

-Lines 131-134 -> perhaps elaborate on the methods used here?

-line 139 -> briefly explain how you got gene age.

Reviewer #3: 

[identifies himself as Nicolas Lartillot]

This manuscript explores the relation between gene age (such as determined by phylostratigraphy) and the rate and patterns of adaptive and non adaptive evolution (such as measured by MacDonald Kreitman approaches), in Drosophila and Arabidopsis. The main results are as follows:

- younger genes undergo higher rates of molecular evolution, both adaptive and non adaptive; this correlation appears to be robust when controlling for multiple confounding factors;

- younger genes tend to make more radical amino-acid changes, and this, both for adaptive and non-adaptive events.

These results are interpreted in terms of an adaptive walk model of gene adaptation and subsequent molecular evolution.

This is a very interesting article. The results presented in it appear to be robust, and the evolutionary hypotheses are stimulating. The article is also well written: very clear, it is a pleasure to read it. I highly recommend it for publication.

I would just have two main comments, one concerning the statistical details of the methods, and another one concerning the interpretation of the results. Then, there are a few very minor comments below at the end.

1. Statistical analyses: strength of evidence versus strength of correlation.

As a general rule, it is useful to discriminate between strength of evidence, on one side, and strength of correlation on the other side. In a typical statistical analysis, information is given separately about the two aspects (typically, a p-value for the strength of evidence, and then a correlation coefficient for the strength of correlation). Strength of evidence scales with sample size (p-values converge to 0 for larger data samples), whereas strength of correlation is a population-level property (correlation coefficients don't systematically increase, they are just more accurately estimated, with larger data samples).

In the present case, however, it seems to me that there is a latent confusion between these two aspects. For instance, the kendall's tau values presented in table 1 are correlation coefficients, so at first sight, one would be tempted to interpret them as a measure of the strength of correlation between gene age and, e.g. the rate of adaptive evolution. Sentences such as 'the effect of gene age prevailed for all estimates in the two species (omega: tau = 0.929, p =1.30e-03)' implicitly convey this message that tau is giving a measure of the strength of correlation, while p is giving a measure of the strength of evidence for this effect size.

However, these tau values are very close to 1, some of them are even essentially equal to 1. This surprised me at first sight. But then I realized that, given the design of the experiment, this is just a consequence of sample size: these tau values measure the correlation between bin median age and bin *mean* effect. The mean effect of a given bin is an average over many genes, and thus much of the variance in gene adaptive or non-adaptive rate has already been factored out at that step. Ultimately, these tau values should all be equal to 1 for very large numbers of genes per bin, which suggests that they should certainly not be taken as a measure of the intrinsic, population-level, strength of the correlation between gene age and other quantities such as omega_a. They cannot be intrinsic, since they scale with sample size (or, here, with bin size).

Similarly, the confidence intervals displayed on the figures such as figure 1 are obtained by bootstrapping the genes within each bin. But then this means that their width should shrink as 1/sqrt(n), with n the number of genes per bin. So, again, the confidence regions displayed on the figures represent the strength of evidence for the correlation patterns, but not directly the strength of the correlation itself: again, they scale with sample size.

Conversely, I don't see any statistical quantity across the article, whether on the text, tables, or figures, that could be taken as a measure of the intrinsic strength of correlation at the gene level: basically, a measure of how much the age of a randomly chosen gene can predict the evolutionary/adaptive rate of this gene (typically, how much of the variance is explained by the covariate). This is missing a lot, I think.

In this respect, there is a point in the discussion:

"we observe that young genes present a 25-fold higher rate of adaptation than older genes in Drosophila species and around 30-fold higher in Arabidopsis. "

-> this sounds like a measure of effect size; but again, it is only about the mean for a given age class; also it is related to the slope of the regression of omega versus gene age, but not to the strength of the correlation. If we take the linear regression case, which is simpler to understand, you can have a steep relation between Y and X (i.e. mean[Y|X=x_highest] can be much higher than mean[Y|X=x_lowest], while still having a weak correlation (var[Y | X] still large, compared to the variance of mean[Y|X] over X).

And thus, I was wondering if the Authors could think a bit about this point and clarify and, perhaps even, enrich, this aspect of their statistical analyses.

- clarify: clearly say, in the main text, or figures, or table legends, whenever a correlation coeff or a confidence interval scales with sample size (possibly indirectly, i.e. by playing with the number of bins, discretization scheme, etc). perhaps expand a bit on the fact that all this does not really measure the intrinsic strength of the correlation between gene age and gene rate of evolution;

- enrich: if possible, give some meaningful measure of the intrinsic gene-level strength of correlation between gene age and molecular evolutionary rates, or proportion of variance explained.

I think one way to estimate the proportion of variance explained would be to compare the bootstrap variance obtained in the experiments done here, with the same bootstrap variance but in a control experiment where genes have been randomly reshuffled across bins, while keeping the same number of genes per bin (thus erasing all information about gene age). If gene age is a good predictor, then the first variance should be smaller than the second, and 1 - v1/v2 should be a measure of the percentage of variance explained by gene age.

Another quicker way would be as follows: assuming that omega_a (or omega_na) is an additive property, such that the mean omega_a for a set of genes is just, conceptually, the mean of the n gene-specific omega_a's, then it would make sense to just inflate the bootstrap variance estimates by n; this should give a rough estimate of the intra-class (i.e. gene-level) true variance of the effect being measured. And then it is relatively simple to compare this intra-class variance (averaged over all bins) with the inter-class variance of the means. One problem is that genes are of varying length, so one should perhaps inflate the variances, not by the true but by the effective number of genes: n_eff = (sum_i L_i)^2 / (sum_i L_i^2), where L_i is the length of gene i.

There are probably better approaches than those suggested here. Of note, I don't think it is a problem if gene age turns out to explain a small proportion of the total variance. Molecular evolutionary patterns are always rather subtle, so one should not consider this possible outcome as a weakness in itself. But it is just that it would be nice to have at least some hint, some quantitative evaluation, or some discussion, about this in the manuscript. In any case, it is important to clarify and to avoid any misunderstanding about the meaning of the statistical measures of strength of signal that are presented.

2. Interpretation of the results.

The interpretation of the results in terms of adaptive walk is definitely an interesting one. However, I could think of other interpretations. For instance: 

Less constrained genes are more easily lost: they can more easily accommodate mis-sense mutations, so they can probably also more easily accommodate non-sense mutations. Therefore, on average, less constrained genes are younger (because the older ones have been lost). In addition, since they accept a larger number of mutations that are otherwise not too deleterious for the folding and for the primary function of the protein, less constrained genes also represent a bigger mutation target for serendipitous adaptations. So less constrained proteins show a higher rate of adaptive evolution. And thus, younger genes, which are enriched in less constrained genes, show a higher rate of both non-adaptive and adaptive evolution. Of note, this mutational target size argument also explains why omega_na and omega_a are correlated, irrespective of gene age. Also, note that the alternative interpretation just suggested is based on a stationary scenario. In contrast, the adaptive walk idea is fundamentally non-stationary.

To be clear: this is not to dismiss the interpretation proposed by the Authors. But it is just that I found the manuscript perhaps a bit too exclusively oriented toward one single 'story' for explaining the pattern, and this at the cost of a broader - and richer - discussion about what could be responsible for these observations. Perhaps, in their discussion, the Authors could give some hints as to other possible interpretations; also, they could give some suggestions as to how one could, in the future, discriminate between these alternative explanations. For instance, since the interpretation proposed by the Author is deeply committed to a non-stationary pattern, it should be detectable by estimating variation in dN/dS across a phylogenetic tree: analyses along those lines would definitely be an interesting perspective, as a way to discriminate between stationary or non-stationary explanations of these findings more generally.

Minor points:

The overall effect of gene age on omega_a and omega_na in each co-factor was assessed by...

-> not clear what is meant by overall effect in each co-factor. perhaps rephrase?

 To do so, we first assessed the correlation of gene age with the rates of molecular evolution in distinct categories of genes, according to a putative confounding factor.

-> not totally clear, phrased like this. Am I correct to understand this: We first sorted genes into classes, according to a putative confounding factor, and then assessed the correlation of gene age with the rates of molecular evolution within each class?

page 5, lines 59: tau = -8.48 ??

not clear why there are two p-values and two tau values. perhaps be more explicit.

- 'controls for confounding effects are only considering two categories (low and high)'

is this control sufficiently tight? Isn't there still some gene age / gene length (or other confounding factor) stratification within each class? In fact, this point could be tested internally: do you still see a significant correlation between gene age and e.g. gene length within the high or within the low class ? An alternative control would have relied on a bins of differing gene ages that are matched for their underlying distribution for gene length (or for any other counfounding factor), although it is not clear to me whether it would be easy to do such matched subsampling in the present case while still guaranteeing sufficiently large sample size within each bin. Of note, the Authors are also using an alternative approach, using linear regression, based on Huang, 2021, which makes their analysis much less dependent on this single control experiment.

- 'We first examined which functions are encoded by young genes in A. thaliana and D. melanogaster...'

-> why only young genes? Wouldn't that make sense to contrast young versus old ? Exactly like the correlation between gene length and gene age was verified before controlling for gene length, earlier in the manuscript, wouldn't that make sense here to test whether some functions are over- or under-represented in young genes versus old genes, before trying to control for this?

- 'When looking at omega_na, our analyses revealed a strong influence of gene age in most functions analysed in both species, where young genes present higher rates of non-adaptive substitutions.'

-> phrasing is slightly ambiguous. Strong influence of gene age on omega_na within most functional classes?

---

## [Decision Letter · Decision Letter 2]

7 Jul 2022

Dear Dr Moutinho,

Thank you for your patience while we considered your revised manuscript "Testing the adaptive walk model of gene evolution" for publication as a Research Article at PLOS Biology. This revised version of your manuscript has been evaluated by the PLOS Biology editors, the Academic Editor and the original reviewers.

Based on the reviews, we are likely to accept this manuscript for publication, provided you satisfactorily address the remaining points raised by the reviewers and the following data and other policy-related requests.

IMPORTANT:

a) Please make your title more explicit and declarative. We suggest "Data from Arabidopsis and Drosophila provide strong evidence for the adaptive walk model of gene evolution" or "Strong evidence for the adaptive walk model of gene evolution in both Drosophila and Arabidopsis."

b) Please attend to the remaining requests from reviewer #1. 

c) Please address my Data Policy requests below; specifically, we need you to supply the numerical values underlying Figs 2ABCDEF, 3ABCD, 4ABCDEFGH, 5AB, S1, S2, S3AB, S4AB, S5ABCDEF, S6AB, S7ABC, S8. I note that your Github deposition currently contains code for the simulations, but please also add the output data. In addition, we need a citeable, permanent record of the data, e.g. in Zenodo, Dryad etc.

d) Please also cite the location of the data clearly in each Fig legend, e.g. “The data and code needed to generate this Figure can be found in https://gitlab.gwdg.de/molsysevol/supplementarydata_geneage and https://zenodo.org/record/XXXXXX.”

We expect to receive your revised manuscript within two weeks. 

*Published Peer Review History*

*Press*

Sincerely,

Roli Roberts

Roland Roberts, PhD

Senior Editor,

rroberts@plos.org,

PLOS Biology

DATA POLICY:

Regardless of the method selected, please ensure that you provide the individual numerical values that underlie the summary data displayed in the following figure panels as they are essential for readers to assess your analysis and to reproduce it: Figs 2ABCDEF, 3ABCD, 4ABCDEFGH, 5AB, S1, S2, S3AB, S4AB, S5ABCDEF, S6AB, S7ABC, S8. NOTE: the numerical data provided should include all replicates AND the way in which the plotted mean and errors were derived (it should not present only the mean/average values).

DATA NOT SHOWN?

REVIEWERS' COMMENTS:

Reviewer #1:

The authors have addressed my previous concerns about the analyses and have vastly improved the manuscript.

I just have a few minor comments to help improve presentation:

1) Lines 245-250: This part was a little unclear. I think a sentence or two are needed toward the beginning of this paragraph giving an overview of what the authors are doing. My understanding is that the authors wish to control for the effect of protein function as a potential confounder on the relationship between omega and gene age. So, they split the genes into different functional categories based on GO terms. Then, within a given GO term, the genes were further divided into different age categories. Grapes was then used to estimate omega. Consider giving an overview like this before delving into the details of how the groups were assigned, etc.

2) Figure S7: The caption here is a little unclear still. Ga is called the proportion of adaptive substitutions and later it's called the "Grantham's distance values" (when referring to the regression line). I think these are the same quantities, but it would be cleaner to use one term in the figure.

3) Methods, section starting on line 520: While this is a little clearer than the previous version of the manuscript, I still think it's hard to follow. I might suggest re-ordering different parts. Specifically, I would first talk about the Grantham distances (like what is done up to line 526). Then, I would suggest talking about estimating omega and Grapes next (ie I would move the parts in lines 539-550 earlier). Lastly, I would talk about the details of inferring the proportion of adaptive nonsynonymous substitutions using the method of Bergman and Eyre-Walker. Right now, it's hard to know why that method is used and what it's being used for. Explicitly explaining that too would make the manuscript clearer here.

Reviewer #2:

The authors have sufficiently addressed my concerns.

Reviewer #3:

The Authors have addressed my comments. I find the experiment controlling for omega_na particularly interesting, in the idea to try to discriminate between the adaptive walk hypothesis and the static model. I still think that more direct tests of the adaptive walk could be considered, e.g. one should see a trend for omega over time. But that's left for future work..

congrats for this beautiful manuscript.

---

## [Editor Report · Decision Letter 3]

1 Aug 2022

Dear Dr Moutinho,

Thank you for the submission of your revised Research Article "Strong evidence for the adaptive walk model of gene evolution in Drosophila and Arabidopsis" for publication in PLOS Biology. On behalf of my colleagues and the Academic Editor, Claudia Bank, I'm pleased to say that we can in principle accept your manuscript for publication, provided you address any remaining formatting and reporting issues. These will be detailed in an email you should receive within 2-3 business days from our colleagues in the journal operations team; no action is required from you until then. Please note that we will not be able to formally accept your manuscript and schedule it for publication until you have completed any requested changes.

Sincerely, 

Roli Roberts

Senior Editor

PLOS Biology

rroberts@plos.org